# Assessment of Seawater Intrusion in Coastal Aquifers Using Multivariate Statistical Analyses and Hydrochemical Facies Evolution-Based Model

**DOI:** 10.3390/ijerph19010155

**Published:** 2021-12-23

**Authors:** Soumaya Hajji, Nabila Allouche, Salem Bouri, Awad M. Aljuaid, Wafik Hachicha

**Affiliations:** 1Laboratory of Water, Energy and Environment, National School of Engineers of Sfax, University of Sfax, B.P. 1173, Sfax 3083, Tunisia; soumaya.hajji@fss.usf.tn (S.H.); nabila.allouche@gmail.com (N.A.); salem.bouri@fss.usf.tn (S.B.); 2Department of Industrial Engineering, College of Engineering, Taif University, P.O. Box 11099, Taif 21944, Saudi Arabia; amjuaid@tu.edu.sa

**Keywords:** groundwater quality, coastal aquifer, hydrochemistry, water resource sustainability, seawater intrusion, principal component analysis (PCA), hydrochemical facies evolution (HFE) model, geographical information system (GIS), Sfax, Tunisia

## Abstract

Groundwater (GW) studies have been conducted worldwide with regard to several pressures, including climate change, seawater intrusion, and water overexploitation. GW quality is a very important sector for several countries in the world, in particular for Tunisia. The shallow coastal aquifer of Sfax (located in Tunisia) was found to be under the combined conditions of continuous drop in GW and further deterioration of the groundwater quality (GWQ). This study was conducted to identify the processes that control GWQ mainly in relation to mineralization sources in the shallow Sfax coastal aquifer. To perform this task, 37 wells are considered. Data include 10 physico-chemical properties of groundwater analyzed in water samples: pH, EC, calcium (Ca), sodium (Na), magnesium (Mg), potassium (K), chloride (Cl), sulfate (SO_4_), bicarbonate (HCO_3_), and nitrate (NO_3_), i.e., investigation was based on a database of 370 observations. Principal component analysis (PCA) and hydrochemical facies evolution (HFE) were conducted to extract the main factors affecting GW chemistry. The results obtained using the PCA model show that GWQ is mainly controlled by either natural factors (rock–water interactions) or anthropogenic ones (agricultural and domestic activities). Indeed, the GW overexploitation generated not only the GWQ degradation but also the SWI. The inverse distance weighted (IDW) method, integrated in a geographic information system (GIS), is employed to achieve spatial mapping of seawater intrusion locations. Hydrochemical facies evolution (HFE) results corroborate the seawater intrusion and its spatial distribution. Furthermore, the mixing ratio showed that Jebeniana and Chaffar–Mahares localities are characterized by high SWI hazard. This research should be done to better manage GW resources and help to develop a suitable plan for the exploitation and protection of water resources.

## 1. Introduction

Generally, coastal aquifers have been exhibiting over-abstraction of groundwater (GW) to face the demand of various human activities such as agricultural and urban activities. Particularly, population and economic growth contribute to a more intensive use of land and a greater pressure on natural resources and ecosystems. This in turn increases the potential threat to the quantity and quality of groundwater [1,2,3,4,5]. Many previous studies based on multivariate analysis such as principal component analysis (PCA), hierarchical cluster analysis (HCA), multiple correspondence analysis (MCA), and geo-statistics demonstrate that the variation of GW quality (GWQ) along the coastline is tightly controlled by both natural and anthropogenic factors: natural factors may be weathering, ion exchange, and rock–water interaction, while anthropogenic factors are represented by domestic, agricultural, and industrial activities [6,7,8,9,10,11,12,13,14,15].

In addition, coastal aquifers are subject to the inflow of seawater, which represents the main limitation for groundwater uses [16,17]. Saltwater intrusion can be considered a very serious hazard to coastal areas that threatens fresh water supplies needed for livelihood [18]. It is plausible to point out that a partial intrusion of seawater into coastal aquifers can take place naturally. It depends on the geological conditions of the reservoirs and the variations in sea level due to global climate changes [19]. In this case, the amplification of the phenomenon may be caused by the excessive use of freshwater [20]. Overcoming this hazard is a major challenge for groundwater management and sustainability. To perform a good diagnostic, an appropriate level of thought about the hydrochemistry pattern is required. 

The present paper focuses on the shallow coastal aquifer of Sfax as a study site. In fact, according to recent studies [21], three models, i.e., fuzzy analytical hierarchy process (FAHP), frequency ratio (FR), and weights of evidence (WOE), are conducted to demonstrate that the high groundwater potential (GWP) is located mainly in the shallow aquifer of the Sfax coastline. Therefore, more attention may be paid to this part for the water resource management plan. During the last decades, the Sfax coastline shallow aquifer was characterized by a continuous lowering of the groundwater level or groundwater degradation and vulnerability of resources to contamination mainly due to seawater intrusion. In fact, a MODFLOW model was conducted to simulate GW-level fluctuations in the shallow aquifer of the Sfax coastline under climate changes and high consumption. This work allows to point out that Jebeniana (in the north) and Chaffar–Mahares (in the south) are the most concerned by the high decrease of the GW level, which makes them the most vulnerable to the hazard of seawater intrusion (SWI) [22]. Furthermore, according to other authors [23], multivariate analyzes and co-kriging tools are used to demonstrate groundwater quality degradation in the Jebeniana locality. This is mainly due to seawater intrusion. Processing with DRASTIC and GALDIT using geographic information system (GIS) tools, study conducted in the study region [24] allowed us to demonstrate that the Agareb–Chaffar–Mahares areas, which are known for rapid population growth and agricultural and industrial development, are characterized by a high-to-medium level of seawater intrusion sensitivity. This is also indicated by other works [25,26,27]. As mentioned above, previous studies tended to focus on aquifer seawater intrusion vulnerability evaluation and vulnerable zone mapping in the Sfax coastline shallow aquifer. This paper aims to assess the SWI hazard by using an integrated approach involving several ionic relationships, hydrochemical facies evolution (HFE) model, GIS, and multivariate statistical analysis. 

## 2. Materials and Methods

### 2.1. Study Area

Geographically, the study area is part of eastern Tunisia and is located between 34°00′–35°30′ N and 9°30′–11°30′ E with a total area of 6848 km^2^. As mentioned in Figure 1, it is limited by the Mediterranean Sea on the east, by the Mahdia region in the north, by the Kairouan and Sidi Bouzid regions in the west, and by the Gabes region in the south. The Sfax region is characterized by a semi-arid to arid Mediterranean climate with irregular temperature and rainfall. The average annual rainfall and temperature are around 229.96 mm and 19.28 °C, respectively [25].

Outcrops of Mio–Pliocene and Quaternary deposits, mainly made up of current and recent alluvial deposits, dominate the lithology of the study area: conglomerates, gravels, sands, and calcareous silts with high permeability [28]. The altitude in the study area rarely exceeds 200 m. The aquifer is logged in the Mio–Pliocene and Quaternary deposits. It is characterized by a thickness ranging between 8 and 60 m with an average of 30 m. The permeability of the aquifer varies from 8 × 10^−6^ to 6.8 × 10^−3^ m/s [24]. The depth of the average GW level of the shallow aquifer varies between 110 and −10 m. The groundwater flow direction is globally from the west to the east toward the Mediterranean Sea, considered as the principal discharge area, and toward secondary outlets, which are the Sebkhas (a smooth flat, often saline plain, sometimes occupied after rain by a shallow lake until the water evaporates).

Calculation and mapping of groundwater-level fluctuations during the period 2008–2017 (Figure 2), under an annual pumping flux of 1.0 × 10^8^ m^3^/year and negative hydraulic budget of −4.1× 10^6^ m^3^, suggest groundwater-level lowering reaching negative values in some specific sites such as Jebeniana and Chaffar–Mahares (−10 m). This explains the groundwater-level drawdown during this period reaching 1.4 m/year corresponding to a lowering of 14 m over the period 2008–2017 (10 years), which is essentially related to the lack of precipitation and excessive and anarchic pumping in these localities [22]. This fact makes the coastline increasingly vulnerable to the threat of marine intrusion. 

Furthermore, groundwater degradation is related to the population growth and the rapid development of different economic sectors (agriculture and industry) and is dependent on a rational water resource management plan. In this context, the aquifer has been the subject of several previous studies, in which more details on geological, hydrogeological, and hydrochemical patterns were given [29,30,31]. 

### 2.2. Sampling and Physico-Chemical Analyses

Thirty-seven groundwater samples were collected, during September 2017, from water supply wells and analyzed in the Sfax, Tunisia, Regional Agricultural Development Committee Laboratory (RCAD-Sfax). Several analytical methods were carried out to assess the chemical composition of the water. Major anions and cations, such as potassium (K^+^), magnesium (Mg^2+^), chloride (Cl^−^), sodium (Na^+^), calcium (Ca^2+^), nitrate (NO_3_^−^), and sulfate (SO_4_^2−^), were analyzed by ion chromatography (Metrohm 850 Professional IC), whereas, the bicarbonate (HCO_3_^−^) level was determined by the potentiometric method (titration with HCl). The electrical conductivity (EC) and pH were measured in situ using an EC meter and a pH meter. The validity of the analyses was checked by calculating the ion balance (IB). All the water analyses results were acceptable (IB < 5%).

### 2.3. Statistical and Geospatial Analyses

PCA is often carried out to identify principal factors affecting groundwater mineralization [11,32,33]. This technique was proven be useful in many previous studies [9,21,24,25,34,35,36]. The dimension-reduction statistical technique has been operated for the case study of the shallow coastal aquifer of Sfax using the SPSS package (version 23). On the other hand, mapping was performed using GIS, which is widely used worldwide for diplaying several phenomenon taking place in aquifers [12,37,38]. By the way, the inverse distance weighted (IDW) and kriging interpolation are the most popular techniques used for mapping in the groundwater field. As many previous studies reported that the IDW interpolation technique is more precise compared to the kriging tool [38,39], the maps in the present work were based on the IDW technique.

### 2.4. Seawater Intrusion Indicators

#### 2.4.1. Chadha’s Diagram

Commonly, sodium and chloride are the dominant ions of seawater/saline water, while calcium and bicarbonate are generally the major ions of fresh water [40]. Therefore, high levels of Na and Cl ions in coastal groundwater may indicate a significant effect of seawater mixing and the occurrence of saline water [41], while considerable amounts of HCO_3_ and Ca reflect mainly the contribution of the water–rock interaction. In this context, the Chadha’s diagram [41] that is obtained by plotting (Ca + Mg) − (Na + K) vs. HCO_3_ − (SO_4_ + Cl), allows to identify the origin of salinization in groundwater. The obtained plot is divided into four fields: (1) recharge water, (2) reverse ion exchange water, (3) seawater effect, and (4) base ion exchange water [5].

#### 2.4.2. Correlation between Hydrochemical Parameters

Several diagrams have been used worldwide to represent the relationship between the major element and other parameters such as EC. Indeed, the plot visualized by the relationship between Cl and Cl/HCO_3_ ratio allows to characterize the origin of salinity in groundwater and, in particular, to assess seawater intrusion in coastal areas [42]. Furthermore, Cl/HCO_3_ ratio is considered a good indicator of salinization by seawater intrusion [5]. Therefore, high values of Cl and Cl/HCO_3_ in the graphic indicate strong seawater intrusion effect. Furthermore, the Ca/Mg ratio is considered among the most significant natural tracers of seawater intrusion in coastal aquifers [43,44]. In fact, the Ca/Mg ratio decreases depending on the proportion of seawater in the mixture of freshwater/seawater. On the other hand, the SO_4_/Cl ratio can be considered as a natural tracer of marine intrusion phenomenon in coastal aquifers. In fact, its variation is inversely proportional to the mixture with seawater [45,46].

The correlation between Cl and EC has been used worldwide to highlight the effect of seawater intrusion. Samples characterized by simultaneously high values of Cl and EC are the most affected by the marine intrusion effect.

### 2.5. Hydrochemical Facies Evolution (HFE) Diagram

Based on the composition of seawater cations/anions (Table 1), tracing the HFE diagram can be powerful to check the SWI locations and the percentage of mixing of freshwater/seawater. In fact, this multirectangular diagram was drawn to highlight the intrusion of seawater by interpreting the mixing of seawater and freshwater mainly in coastal aquifers [47,48,49]. Based on seawater cation/anion composition, the diagram allows one to (1) calculate the percentage of cations and anions in GW, (2) identify the sequencing stages of seawater intrusion to freshwater sources as (i) direct ion exchange, (ii) mixing, and (iii) reverse cation exchange, and (3) determine hydrochemical facies in different boreholes. More details are introduced in previous studies [50].

## 3. Results and Discussion

### 3.1. Statistical Results

The database is given in Table 1 as minimum, maximum, average, and standard deviation values for physicochemical elements. For carring out PCA, 10 variables were taken into considreation: Na, Mg, Ca, K, Cl, SO_4_, HCO_3_, NO_3_, pH, and EC. The results obtained show a total explained variance (σ^2^) equal to 81.4% (after varimax rotation). Moreover, the Kaiser–Meyer–Olkin Measure of Sampling Adequacy (KMO) was equal to 0.52, and according to the Kaiser criterion, eigenvalues >1. Three principal components were retained: PC1 (σ^2^ = 41.3), PC2 (σ^2^ = 23.8), and PC3 (σ^2^ = 16.35).

As expected, PC1 is strongly supported by EC (0.917) and some major ions (Na, Cl, Ca, SO_4_, Mg) (Table 2). It may be related to natural effects such as evaporation, rock–water interactions, and probably the seawater intrusion effect. To prove this hypothesis, some indicators such as Chadha’s diagram, some binary diagrams, and the hydrochemical facies evolution diagram were obtained. PC2 loading NO_3_ and negatively loading pH represents the influence of agricultural activities and alkalinity as the major character of mineral fertilizers. PC3 having the lower eigenvalue (1.428) showed 16.4% of the total variance. It showed significant factor loading of HCO_3_ and K (Figure 3). This could be related to the influences of domestic activities on groundwater resources [51,52] but could also be associated with direct recharge of precipitation as a result of the oxidation effecting coastal aquifers. The positive correlation between HCO_3_ concentration and GW level would seem to explain this phenomenon of HCO_3_ enrichment due to direct recharge from precipitation (Figure 4). The same observation has been made in a previous study for SO_4_ element [53].

As the effects of agricultural and domestics activities on Sfax unconfined groundwater quality have been revealed by different authors [23,24,25,26,27], we only focus in the present paper on marine intrusion given its effect on the degradation of the groundwater quality.

### 3.2. Water Chemical Features

Several hydrochemical studies conducted on the Sfax aquifer indicated that groundwater mineralization is controlled by many factors such as evaporation, precipitation, and dissolution, which are considered to be more expressed in shallow aquifers. These phenomena were also observed in a case study of the coastal aquifers of southern India [50] and in the case of the coastal Djeffara aquifer, southeastern Tunisia [54]. Indeed, chemical analyses conducted on the Sfax coastal shallow aquifer indicate that the groundwater is characterized by NaCl–CaSO_4_ facies, which denotes an origin of mineralization from the dissolution of evaporitic minerals such as halite and gypsum–anhydrite.

Furthermore, hydrochemical models carried out to understand the mechanisms of salinization variations in groundwater [55] demonstrate that basic ion exchange mechanisms could play a role in the mineralization of water in the shallow aquifer of the Sfax coastal. Indeed, the majority of analytical results confirm the participation of Na, Ca, and Mg in the ion exchange. Furthermore, the dominance of Ca + Mg over HCO_3_ + SO_4_ indicates the presence of reverse ion exchange. Furthermore, high concentrations of Na as compared to Ca and Mg in some locations can result from saline water intrusion [56]. Based on previous studies taking place in the Sfax coastal shallow aquifer, the extent and the weight of seawater intrusion risk are parameters that need to be determined. For this task, several indicators need to be considered.

### 3.3. Seawater Intrusion Indicators

According to Chadha’s diagram (Figure 5), the boreholes that are affected by the seawater intrusion (samples no. 3, 4, 18, 19, 22, 24, 26, 27, 28, 37) are located in the coastline of the shallow aquifer of Sfax, corresponding to the Chaffar–Mahares–Agareb–Jebeniana and Skhira localities (Figure 1).

Based on the Cl vs. Cl/HCO_3_ ratio, groundwater can be classified into unaffected (<0.5), slightly or moderately affected (0.6–6.6), and strongly affected (>6.6) by the salinization process (more detail in [5]). The vertical dotted line, corresponding to the Cl concentration of 65 mg/L, indicates the limit after which a strong seawater effect was recorded. In the present study, this relationship (Figure 6) shows that groundwater in the Sfax shallow coastal aquifer is strongly affected by seawater intrusion in the Jebeniana locality (samples no. 32 and 34) and the Chaffar–Mahares locality (samples no. 18 and 19).

The Cl/HCO_3_ spatial distribution generated using GIS tools according to the IDW technique (Figure 7) showed the highest values in the coastline of the study area, especially in Jebeniana and Chaffar–Mahares–Agareb and slightly high values in Skhira.

#### Correlation between Cl and Ca/Mg Ratio

The plot of Ca/Mg vs. Cl showed the marine effect in samples no. 8, 18, 19, 22, 32, and 34 (Figure 8), whose ratio is close to seawater and having high Cl concentrations. The high Ca/Mg ratio registered in the samples located in inland areas may be related to another effect, such as water–rock interaction and ion exchange.

On the other hand, based on the SO_4_/Cl ratio, it can be noticed that the low ratio values indicated in many wells (Figure 9) contribute to highlight saltwater/freshwater mixing in some parts of the aquifer (samples no. 3, 8, 18, 19, 22, 27, 28, 32, 34), located in the Jebeniana, Chaffar–Mahares, and Skhira regions. Enrichment of SO_4_ in other localities may be related to either rock interaction or base ion exchange.

Furthermore, the geospatial distribution obtained by using the IDW interpolation technique under GIS tool (Figure 10) confirmed the obtained results.

Indeed, Jebenina, Chaffar–Mahares, and Skhira regions, located along the coastline of the study area, are characterized by the lowest values of SO_4_/Cl ratio showing the risk due to effect of the seawater intrusion.

Furthermore, the correlation between Cl and CE (Figure 11) was used not only to show the strong correlation between the two parameters (R^2^ = 0.8257) in all samples but also to express the evolution from freshwater to seawater and the interaction between the two types of water. The pole influenced by marine intrusion effect is characterized by high rates of both Cl and EC. This is also observed mainly in Jebeniana and Chaffar–Mahares and slightly in Skhira localities.

### 3.4. Hydrochemical Facies Evolution (HFE) Diagram Results

The HFE is achieved using a code in an EXCEL file with a Macro (Figure 12). It reveals that groundwater in coastal shallow aquifer of Sfax mainly consists of Na-Cl and Ca-SO_4_ water types. Moreover, the majority of the samples are situated on the left of the mixing line, indicating freshwater under the influence of direct cation exchange. As noted in Table 3, the Ca-SO_4_ mix facies is observed in 27% of the samples (10 wells). The same percentage was assigned to the mix Na-Cl facies. Therefore, 54% of the samples was influenced by direct cation exchange and 46% by the seawater intrusion effect that is especially observed in the Jebeniana (samples no. 32, 34, 36) and Chaffar (sample no. 25) regions. This corroborates the results obtained by vulnerability assessment studies conducted in the shallow aquifer of Sfax [25,26,27]. These results confront those obtained other authors [22], in that they demonstrate that Jebeniana and Chaffar regions are characterized by the higher average of groundwater-level drawdown under climate change scenarios and are the most threatened by seawater intrusion hazard in the Sfax region.

### 3.5. Mixture with Seawater Estimation

The calculation of mixing rate in aquifers is usually based on geochemical tracer models, which may be the origin of uncertainty. This is due to chemical reactions that occur in groundwater. To overcome this problem, conservative tracers may be used [57,58]. In the present case study, chloride (Cl) is considered a conservative tracer, and it was used for the assessment of the mixing rate (Rmix) [44,59] between freshwater and seawater. The calculation of Rmix is based on the mass conservation (Equation (1)).
(1)$$\mathrm{Rmix}=\frac{{\mathrm{Cl}}_{\mathrm{sample}}{-\mathrm{Cl}}_{\mathrm{freshwater}}}{{\mathrm{Cl}}_{\mathrm{seawater}}{-\mathrm{Cl}}_{\mathrm{freshwater}}}\times 100,$$
where Cl_sample_ denotes the chloride concentration in the water sample, Cl_freshwater_ denotes the chloride concentration in freshwater (as the average chloride concentration of samples having electric conductivity <1000 μS/cm = 46 mg/L) and Cl_seawater_: chloride concentration in seawater (=19,000 mg/L) 

The calculated mixing rate (Rmix) ranges from 0% to 18.12% (Figure 13). The highest value belongs to the Jebeniana locality, whereas the lowest belongs to the extreme inland area of the Mahares locality. High values are obviously recorded along the coastline. The degree of risk of seawater intrusion phenomenon is evaluated according to the Rmix values mentioned in Table 4. As anticipated, Jebeniana is characterized by very high to high risk, Mahares–Chaffar denoted high to medium risk, and Skhira showed medium risk of seawater intrusion, while the inland area revealed a low degree of contamination risk. This latter result is related to the piezometric level fall, the low renewal rate by precipitation decrease, and the sea level elevation.

## 4. Conclusions

The shallow Sfax coastal aquifer is considered to be stressed due to continuous water level drawdown and degradation of water quality. Seawater intrusion risk is the common point between both factors. Indeed, it is, simultaneously, (i) the effect of overexploitation due to population growth and water supply demand increasing, and (ii) the cause of water salinization. Consequently, this area should be considered by the managers in order to minimize groundwater degradation generated mainly by seawater intrusion risk and overexploitation. In the present study, PCA was performed to show that groundwater quality was affected by three factors with PC1 indicating natural sources of contamination including seawater intrusion, and PC2 and PC3 being related to anthropogenic activities. The hypothesis of marine intrusion was checked by some indicators involving Chadha’s diagram, major-ion correlation diagrams, and hydrochemical facies evolution. In addition, it is worth mentioning that the processing of mapping seawater intrusion indicators such as SO_4_/Cl ratio and HCO_3_/Cl ratio using IDW interpolation under the GIS tool was proven useful to visualize the spatial distribution of the affected zones. The contribution of this investigation is not only to highlight the risk locations in the area of survey but also to quantify the percentage of seawater intrusion as the mixing ratio of freshwater/seawater by using the HFE model. Furthermore, the calculation of the freshwater/seawater mixing rate (Rmix) was found helpful to attribute seawater intrusion risk degree ranking for each affected zone. In fact, the Jebeniana locality showed boreholes under risk of seawater intrusion ranging from medium to very high. It depends probably on borehole distance to the coast, while Skhira was still found to be at medium risk. Obviously, great attention must be paid to groundwater along the coastal regions, especially in the Skhira locality, which should be considered by water authorities as a “protected perimeter” from irrational practices such as overexploitation and uncontrolled human activities (agricultural, industrial, and domestic). This should be taken into account for future management studies regarding groundwater quality and suitability for consumption.

## Figures and Tables

**Figure 1 ijerph-19-00155-f001:**
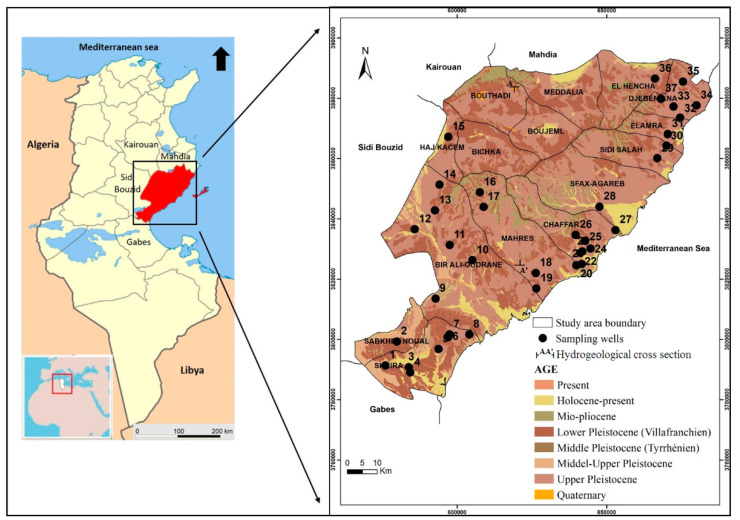
Geological map of the study area.

**Figure 2 ijerph-19-00155-f002:**
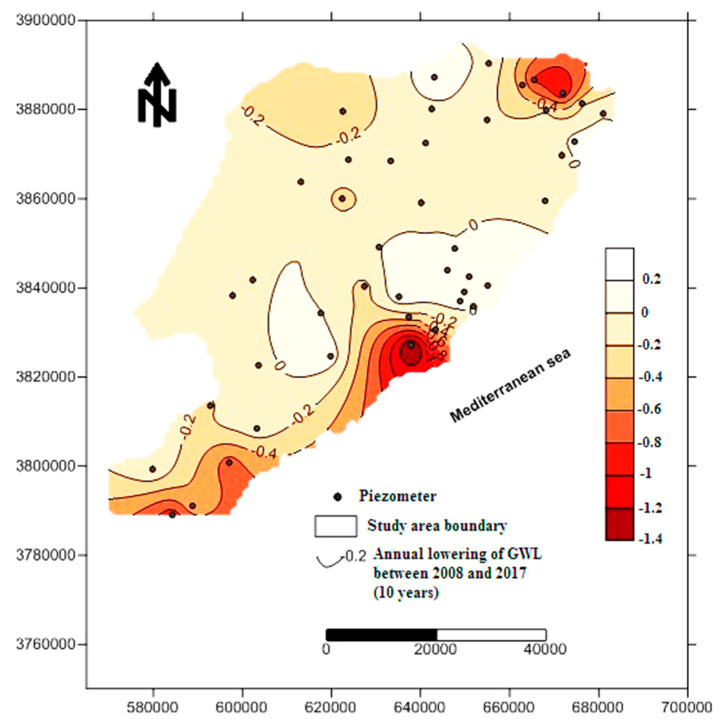
Groundwater-level lowering (m/year in 10 years in the period 2008–2017).

**Figure 3 ijerph-19-00155-f003:**
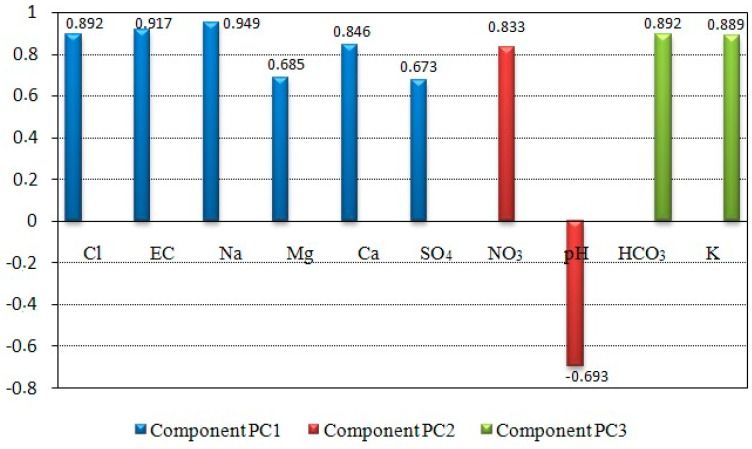
Most relevant PCA result: PC1, PC2, and PC3.

**Figure 4 ijerph-19-00155-f004:**
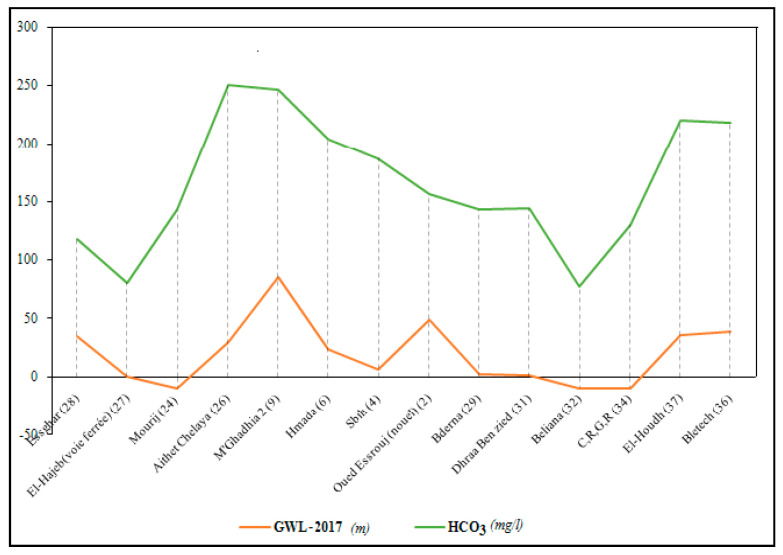
Correlation piezometric on HCO_3_ spatial evolution in 2017 in the coastal shallow aquifer of Sfax.

**Figure 5 ijerph-19-00155-f005:**
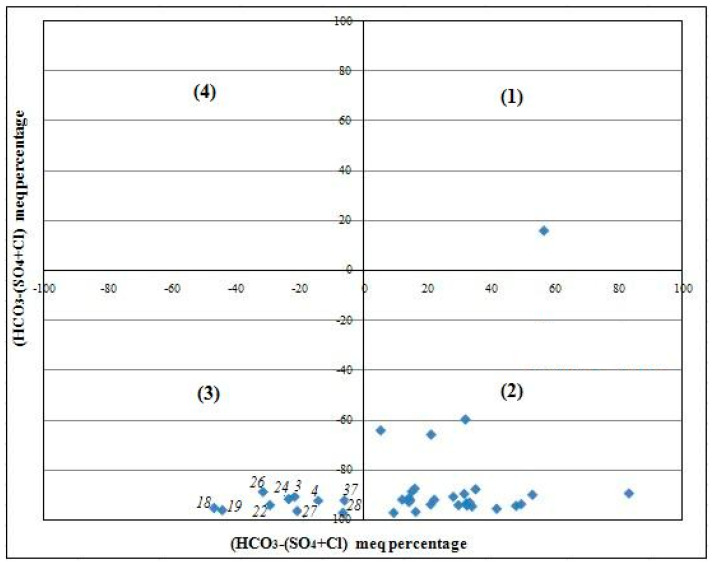
Chadha’s diagram in the Sfax coastal shallow aquifer: (**1**) recharge water, (**2**) reverse ion exchange water, (**3**) sea water effect, (**4**) base ion exchange water.

**Figure 6 ijerph-19-00155-f006:**
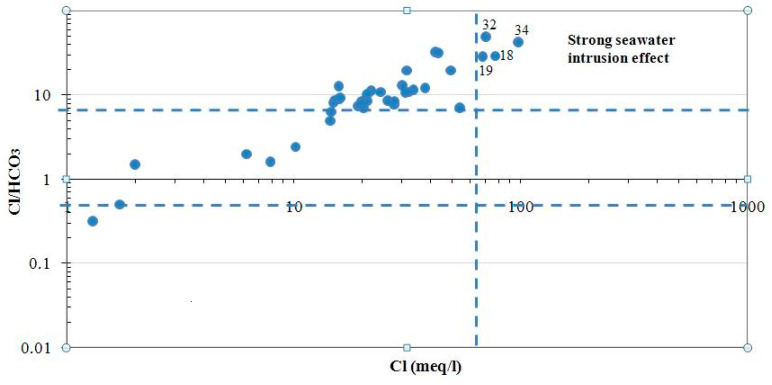
Relationship Cl/HCO_3_ vs. Cl in groundwater of the Sfax coastal shallow aquifer.

**Figure 7 ijerph-19-00155-f007:**
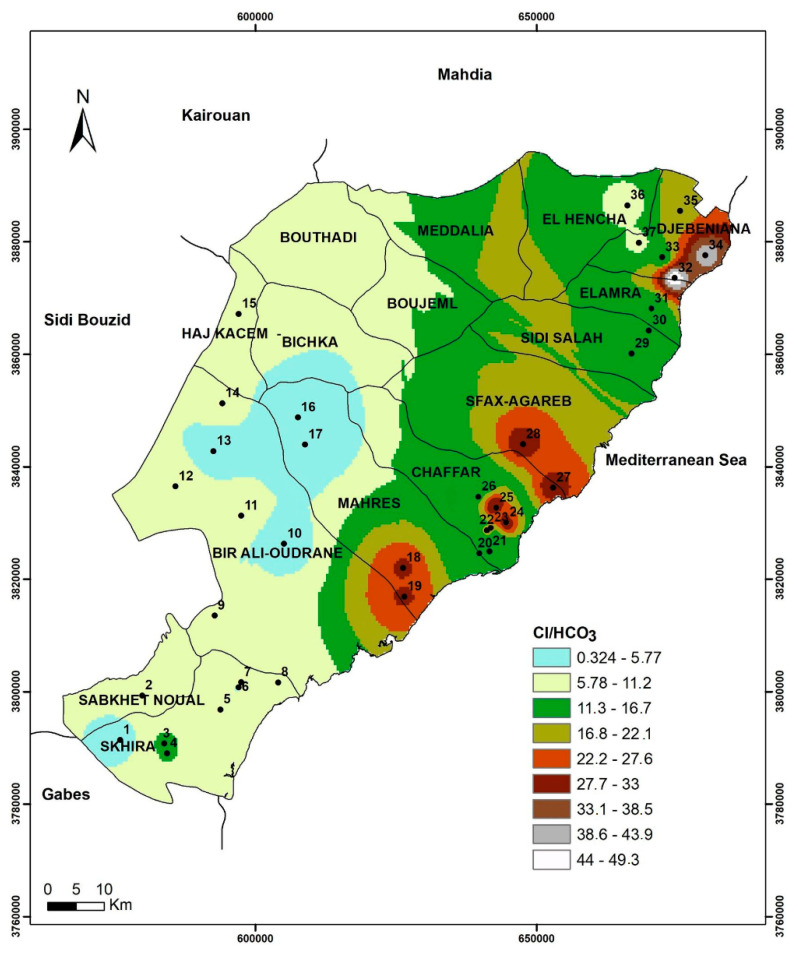
Cl/HCO_3_ spatial distribution in study area.

**Figure 8 ijerph-19-00155-f008:**
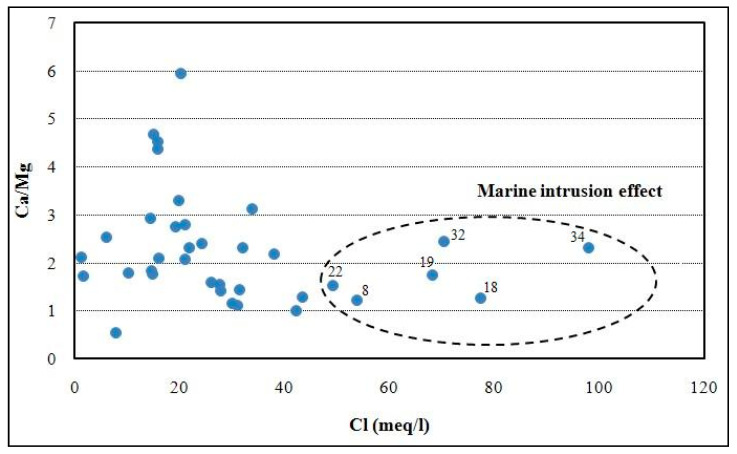
Ca/Mg vs. Cl plot in the study area.

**Figure 9 ijerph-19-00155-f009:**
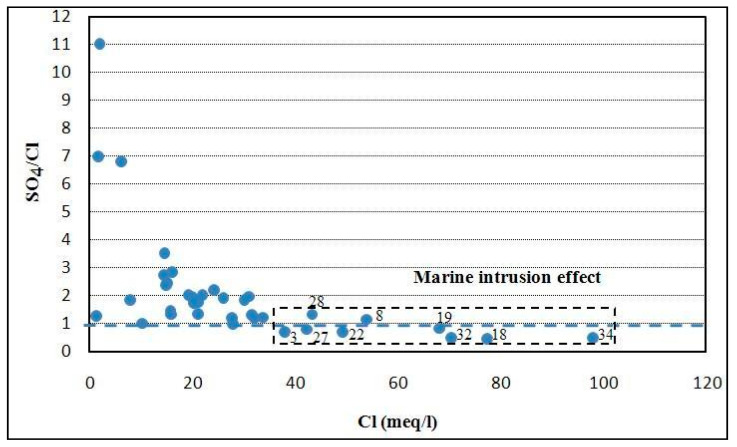
SO_4_/Cl vs. Cl plot in study area.

**Figure 10 ijerph-19-00155-f010:**
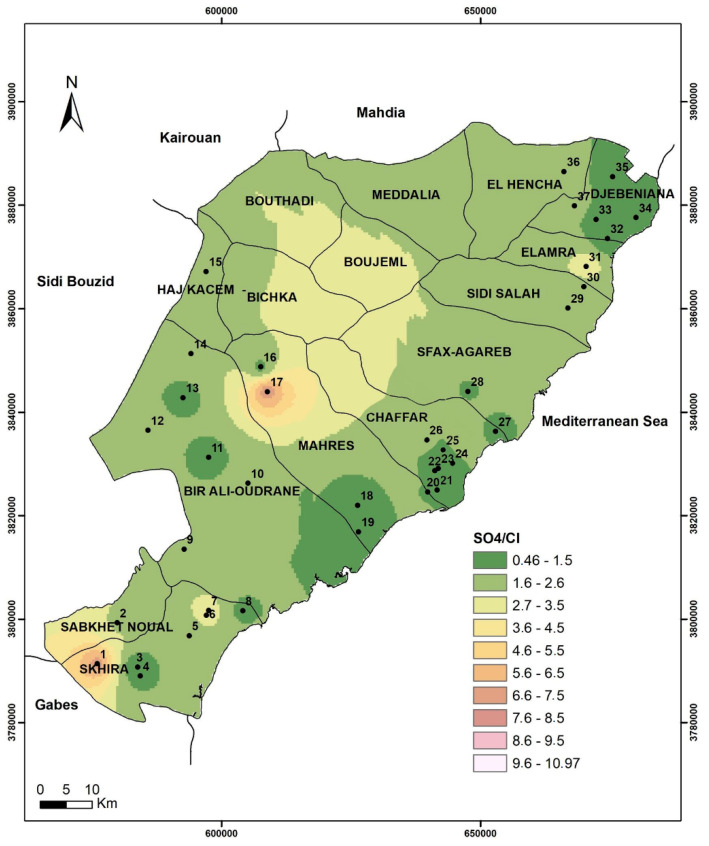
SO_4_/Cl geospatial distribution in the study area.

**Figure 11 ijerph-19-00155-f011:**
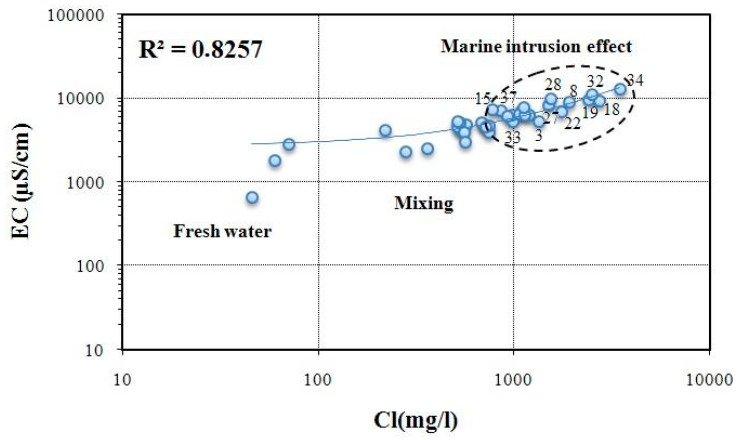
EC vs. Cl plot in the study area.

**Figure 12 ijerph-19-00155-f012:**
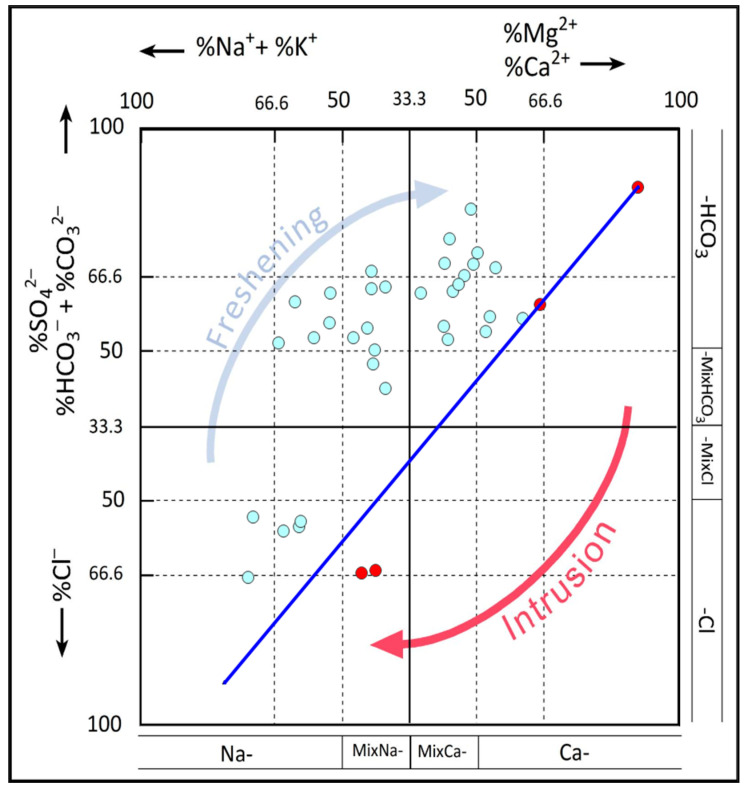
Hydrochemical facies evolution model for the Sfax coastal shallow aquifer (Blue: freshwater, red: water affected by SWI).

**Figure 13 ijerph-19-00155-f013:**
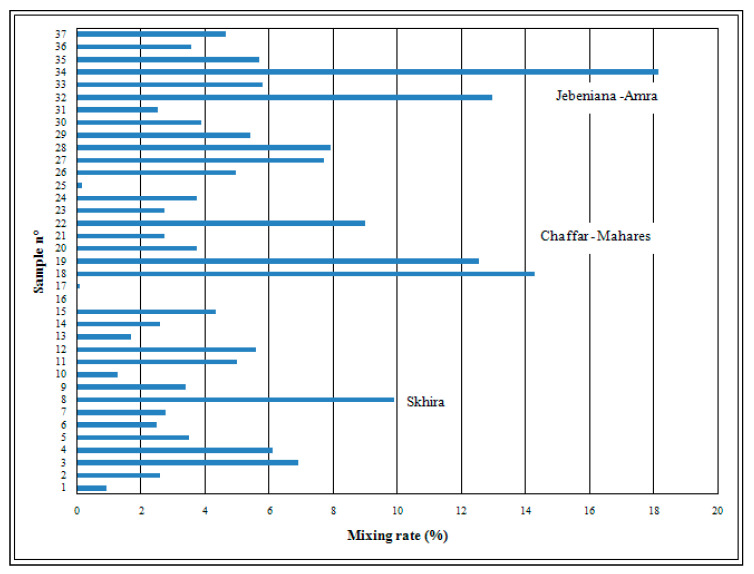
Rmix evaluation and spatial distribution.

**Table 1 ijerph-19-00155-t001:** Descriptive statistics of physicochemical parameters (contents in mg/L).

	*N*	Minimum	Maximum	Mean	Std. Deviation
pH	37	6.33	7.89	7.39	0.352
EC (μS/m)	37	6.6 × 10^2^	1.3 × 10^4^	5.8 × 10^3^	2.6 × 10^3^
Na	37	3.2 × 10	2.3 × 10^3^	7.7 × 10	5.3 × 10^2^
Mg	37	6.50	3.8 × 10^2^	1.6 × 10^2^	9.5 × 10
Ca	37	9.0 × 10	1.4 × 10^3^	5.4 × 10^2^	2.4 × 10^2^
NO_3_	37	4.00	1.6 × 10^2^	4.5 × 10	3.0 × 10
SO_4_	37	8.1 × 10	3.0 × 10^3^	1.8 × 10^3^	7.0 × 10^2^
HCO_3_	37	7.5 × 10	4.7 × 10^2^	1.6 × 10^2^	7.4 × 10
Cl	37	4.6 × 10	3.5 × 10^3^	1.0 × 10^3^	7.8 × 10^2^
K	37	1.70	62.00	14.50	10.61

**Table 2 ijerph-19-00155-t002:** Component matrix using principal component analysis as an extraction method.

	Component		
	PC1	PC2	PC3
pH	−0.176	−0.693	−0.113
EC	0.917	0.354	−0.027
Na	0.949	−0.080	−0.040
Mg	0.685	0.392	0.380
Ca	0.846	0.387	−0.018
SO_4_	0.673	0.412	0.216
NO_3_	0.072	0.833	−0.011
Cl	0.892	0.240	−0.030
HCO_3_	−0.074	−0.172	0.892
K	0.157	0.290	0.889

**Table 3 ijerph-19-00155-t003:** Phases and facies of groundwater samples in the Sfax coastal shallow aquifer.

Borehole No.	Phase	Facies	
		Cations	Anions
1	Fresh	MixCa	SO_4_
2	Fresh	Ca	SO_4_
3	Fresh	Na	Cl
4	Fresh	Na	SO_4_
5	Fresh	MixCa	SO_4_
6	Fresh	MixCa	SO_4_
7	Fresh	Ca	SO_4_
8	Fresh	MixNa	SO_4_
9	Fresh	MixNa	SO_4_
10	Fresh	MixNa	SO_4_
11	Fresh	MixNa	MixSO_4_
12	Fresh	MixNa	SO_4_
13	Fresh	MixNa	MixSO_4_
14	Fresh	MixNa	SO_4_
15	Fresh	MixCa	SO_4_
16	Fresh	Ca	HCO_3_
17	Fresh	MixCa	SO_4_
18	Fresh	Na	Cl
19	Fresh	Na	Cl
20	Fresh	MixNa	SO_4_
21	Fresh	Ca	SO_4_
22	Fresh	Na	Cl
23	Fresh	Ca	SO_4_
24	Fresh	Na	SO_4_
25	Intrus	Ca	SO_4_
26	Fresh	Na	SO_4_
27	Fresh	Na	Cl
28	Fresh	Na	SO_4_
29	Fresh	MixCa	SO_4_
30	Fresh	MixCa	SO_4_
31	Fresh	MixCa	SO_4_
32	Intrus	MixNa	Cl
33	Fresh	MixCa	SO_4_
34	Intrus	MixNa	Cl
35	Fresh	MixCa	SO_4_
36	Intrus	Ca	SO_4_
37	Fresh	Na	SO_4_

**Table 4 ijerph-19-00155-t004:** Spatial ranking of seawater hazard.

Rmix (%)	Boreholes	City	Seawater Intrusion Risk
20–15	34	Jebeniana	Very high
15–10	18, 19, 32	Jebeniana–Mahares– Chaffar	High
10–5	3, 4, 8, 12, 22, 28, 29, 27, 33, 35	Skhira–jebeniana–Agareb–Chaffar	Medium
5–0	The rest of samples	Inland area	Low

## Data Availability

Data are contained within the article.

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
