# Peer review of "Assessment of Seawater Intrusion in Coastal Aquifers Using Multivariate Statistical Analyses and Hydrochemical Facies Evolution-Based Model"

_ijerph, 2021, doi:10.3390/ijerph19010155_

Round 1

Reviewer 1 Report

1. Suggestions:
S1: suppress acronym definition in abstract ( usually they are explicit in current text )

e.g.:
Instead of...
"Abstract: Groundwater (GW) studies have been conducted worldwide by several pressures, includ-
ing climate change, seawater intrusion (SWI), and water overexploitation. GW quality (GWQ) is a..."

write only...
"Abstract: Groundwater studies have been conducted worldwide by several pressures, includ-
ing climate change, seawater intrusion, and water overexploitation. GW quality is a..."
---------------------------------------------------------------------------------------------
S2: in Table 1...
a) organize analytical parameters in physico-chemical determinations ( pH and EC ) and than chemical parameters ( Na, Mg, ... )
b) please refer units for EC...
c) accordingly EURACHEM/CITAC uncertainty representation the right number of significant numbers for estimates are obtained by:
  i) represent uncertainty with only 2 significant digits ( e.g. for EC StDev(EC) = 2612.18 -> should be indicated as StDev(EC) = 2.6e4 )
  ii) represent corresponding central estimate with same number of decimal places ( mean EC = 5798.24 should be rounded to EC = 5.8e4 )

Question about your statistical inference in table...
Q3: since you are dealing with contamination issues... makes sense to evaluate mean value and dispersion in standard deviation across different 37 groundwater wells?

To avoid this problem I suggest...
> since you have only 37 groundwater samples analysed accessed for 10 quality parameters (variables), why not show these values (tabulate) and add (in last rows) corresponding estimates?
> in this way your results may be used to allow other other scientists to follow your results and methodology in deep
---------------------------------------------------------------------------------------------
S3: in sub-section "3.1. Statistical results" 
> correct "...Morrrrrver" to "More ever"
> represent percent of recovered information with 1 decimal place and add the corresponding "%" symbol
---------------------------------------------------------------------------------------------

#############################################################################################
2. Questions:
---------------------------------------------------------------------------------------------
Q1: in sub-section "2.2. Sampling and physic-chemical analyses"
authors refer...
"Atomic Absorption Spectrometer for Nitrate and Nat..."

a) AAS for nitrate determination?  
Please tell me more...  I don't know this analytical method...

b) what is the meaning of "Nat"?
---------------------------------------------------------------------------------------------
Q2: in sub-section "2.2. Sampling and physic-chemical analyses"
"...The electrical conductivity (EC) and pH were measured in situ using an Ec-meter and a pH-meter."
> can you please specify in more detail used equipment ( Provider, model, electrodes... ) for EC and pH measurements?
---------------------------------------------------------------------------------------------
Q3: Table 1
since you are dealing with contamination issues... makes sense to evaluate mean value and dispersion in standard deviation for each 10 measured parameters across different 37 groundwater wells?
---------------------------------------------------------------------------------------------
Q4: in sub-section "3.1. Statistical results"
"The data base is given by Table 1 as minimum, maximum, average, and standard
deviation values for physicochemical elements"
> this comment is problematic...  your PCA analysis was done with a data matrix of X(37x10) (37 samples and 10 variables)
> it's preferable to replace Table 1 by actual results used in PCA
---------------------------------------------------------------------------------------------
Q5: in sub-section "3.1. Statistical results"
> please specify the pre-processing step used in PCA (are you working in respect to covariance or to correlation?)... This will conditioned all PCA results)
---------------------------------------------------------------------------------------------
Q6: in Table 2, what was the criteria to only evidence "loadings" for PC1 to PC3 for some variables?
Why not put all information end evidence in BOLD only higher values?
---------------------------------------------------------------------------------------------

#############################################################################################
3. Problems:
---------------------------------------------------------------------------------------------
P1: in sub-section "2.4. Hydrochemical Facies evolution (HFE) diagram"
> in section "2.2. Sampling and physic-chemical analyses" authors reveal collecting "thirty-seven (37) groundwater samples"
> they accessed 10 quality parameters (variables, see Table 1)
> no replicates?

In table 1 -> performed numerical evaluations (mean and standard deviation)
> is it valid to make standard estimates (use least squares approach) where there are no replicates and outliers 
should be present?
> suggestion: represent in Table 1 actual measured values and than present "robust" estimates (median based)
---------------------------------------------------------------------------------------------
P2: I don't see the need to represent results from "Table 2" in "Figure 3"
Preferable to represent all "loadings" in a 3-column figure?   
---------------------------------------------------------------------------------------------
P3: instead of "Figure 4" representation...
To better evidence this "correlation" represent in a 2D scatter plot [HCO3] (vertical axis) in respect to
NP (2017) (horizontal axis) for all 37 wells! (instead of only 14 cases)
---------------------------------------------------------------------------------------------
P4: Why not to explore PCA scores plot and compare this analysis with "hydrochemical Facies water characterization"  for seawater intrusion ( e.g. data represented in Figure 5 and others )
---------------------------------------------------------------------------------------------

Author Response

ijerph-1494949

Title:  Assessment of Seawater Intrusion in Coastal Aquifers using Multivariate Analyses and Hydrochemical Facies Evolution-based Model

The authors would like to thank the editor and the anonymous reviewers, whose insightful comments and constructive suggestions helped us to significantly improve the quality of this paper. Every change in the text is colored in red

Reviewer 2 Report

The topic of the article is interesting and deserves publication, as it helps to identify where and which processes control the groundwater quality of a shallow coastal aquifer. Overall, but, according my opinion, could be improved.

The text and diagrams need general revision and sometimes the explanations are not very clear. A deepening of the results with simple and clear explanations is necessary, only in this way the manuscript becomes complete and publishable.

Finally, according my opinion, the conclusions need to be expanded : for example, you should try to highlight the most critical zones, i.e. those that have simultaneously shown a high lowering of the water table, a high risk of salt intrusion with a predominant Na-Cl facies, a high Cl/HCO3 ratio and electrical conductivity, and a low SO4/Cl ratio. There is no comment on the less critical zones (indicate which are the least critical and try to explain why they are critical). There is no comment on the results obtained from the Principal Component Analysis.

It is suggested that this paper can be accepted for publication after moderate major revisions.

Author Response

(The authors gave the same response as above.)

Reviewer 3 Report

This manuscript assessed the level and risk of seawater intrusion in coastal aquifers by statistical and model analyses. Generally, the manuscript is well written and the results are well presented. However, some important information need to be clarified. Specific comments for the potential revision are as follows:

  1. “Seawater intrusion” was abbreviated as “SWI” in line 12 but to “SW” in line 78. In addition, the abbreviation should be defined when it appeared for the first time in line 68.
  2. Why the sampling wells in Fig. 1 are different from those in Fig. 2? The authors mentioned that “thirty-seven (37) groundwater samples were collected” in line 113, where are they?
  3. Figure 2 and Figure 7. There are several numbers outside the border. What are they?
  4. Line 101, please add a space between “10” and “m”.
  5. Line 116, what does “Nat” refer to? Can you give more information of the instrument (model and manufacturer) used to measure the water quality of groundwater?
  6. Delete the full name of PCA in line 122 because it has been defined in line 41.
  7. Figure 6. What does the dash lines stand for?
  8. Why different units were used for chloride in Figure 9 and Figure 11?
  9. Use superscript and subscript where it is needed. Please check throughout the manuscript.
  10. Check the format of references in the list.

Author Response

(The authors gave the same response as above.)

Round 2

Reviewer 1 Report

Dear authors:

Thank you for your changes in text that consolidate your effort in this work.
Well done.

Author Response

Round 2 :

ijerph-1494949

Title:  Assessment of Seawater Intrusion in Coastal Aquifers using Multivariate Analyses and Hydrochemical Facies Evolution-based Model

The authors would like to thank the editor and the anonymous reviewers, whose insightful comments and constructive suggestions helped us to significantly improve the quality of this paper. Every change in the text is colored in red. 

Many thanks 

Reviewer 2 Report

In this second revision, the context of the study was improved overall. In my opinion it can be published but I suggest still some minor corrections to be made.

Round 2

ijerph-1494949

Title:  Assessment of Seawater Intrusion in Coastal Aquifers using Multivariate Analyses and Hydrochemical Facies Evolution-based Model

The authors would like to thank the editor and the anonymous reviewers, whose insightful comments and constructive suggestions helped us to significantly improve the quality of this paper. Every change in the text is colored in red.

Many thanks 
